# Role of community pharmacists in the safe and effective use of complementary and alternative medicine in the Middle East: A scoping review

Farah Naja[1,2], Maiss Ahmad[3], Hibeh Shatila[4], Katia Hazim N. Abu Shihab[1,2], Sally Naalbandian [5], Roba Saqan[2], Hamzah Alzubaidi[2,6], Mohamad Ali Hijazi [7]*

1 Department of Clinical Nutrition and Dietetics, College of Health Sciences, University of Sharjah, Sharjah, United Arab Emirates, 2 Research Institute of Medical & Health Sciences (RIMHS), University of Sharjah, Sharjah, United Arab Emirates, 3 Department of Applied Health Sciences, Public Health Programme, University of Birmingham Dubai, International Academic City, Dubai, United Arab Emirates, 4 Human Nutrition Department, College of Health Sciences, QU Health, Qatar University, Doha, Qatar, 5 Science & Agriculture Library, University Libraries, American University of Beirut, Beirut, Lebanon, 6 Department of Pharmacy Practice and Pharmacotherapeutics, College of Pharmacy, University of Sharjah, Sharjah, United Arab Emirates, 7 Pharmaceutical Sciences Department, Beirut Arab University, Beirut, Lebanon

* m.hijazi@bau.edu.lb

## Abstract

### Background

Community pharmacists (CPs) play a crucial role in promoting safe use of Complementary and Alternative Medicine (CAM). This scoping review examined existing evidence on the role of CPs in CAM in the Middle East, a region harboring one of the fastest growing markets of CAM.

### Methods

A comprehensive search of nine electronic databases was conducted to identify relevant studies published between 2000 and 2025. A total of 155 studies met the inclusion criteria. Key study characteristics were extracted using a standardized data extraction form. Thematic analysis of findings and recommendations was carried out using NVivo software to identify common patterns, themes, and gaps in the literature.

### Results

The studies primarily used quantitative cross-sectional designs and were predominantly led by academic institutions. International and regional collaborations were scarce. Over the past 25 years, there has been a notable rise in the annual number of published papers; however, this upward trend was not reflected in the impact factors of the journals in which these papers appeared. The thematic analysis revealed three main themes: (1) Opportunities, including pharmacists' positive attitudes and

**Data availability statement:** All relevant data are within the manuscript and its Supporting information files.

**Funding:** The author(s) received no specific funding for this work.

**Competing interests:** The authors have declared that no competing interests exist.

willingness to take on active roles in CAM; (2) Challenges, such as knowledge gaps, poor reporting practices, logistical barriers to counseling, weak regulatory oversight, and misinformation driven by media; and (3) Recommendations, targeting improvements in practice, education, and policy.

## Conclusions

The review findings suggest the need for studies to diversify their designs and strengthen cross-sector collaborations. To enhance CPs' role in ensuring the safe use of CAM, coordinated, evidence-based strategies are required. These should integrate professional development, regulatory reform, and targeted public education, ultimately supporting CPs in delivering informed, patient-centered CAM services.

## Introduction

Over the last decade, there has been a worldwide increase in the use of complementary and alternative medicine (CAM) [1,2], in all its forms. The National Center for Complementary and Integrative Health classify CAM as a group of diverse nutritional products, physiological and physical practices and any combination between them [3]. Reports on CAM use from around the world showed prevalence rates as high as 74% among the general populations and 96% among patient populations [4]. Among the factors that contribute to the popularity of CAM are the consumers' dissatisfaction with the efficacy or safety of conventional medicine [5], their perception that CAM products or procedures are safe and made of natural materials, their individual desire to take an active role in their healthcare decisions, cultural beliefs, social networks and the media's influence [1,6,7].

The widespread use of CAM and its growing popularity could in many instances threaten public health, specifically in the absence of rigorous clinical evidence about the safety or efficacy of some CAM modalities [7–9]. Adverse outcomes associated with CAM may occur as direct and/or indirect effects of their use. The direct toxicity of CAM products comes from possible overdose or natural characteristics of their constituents, such as the case of excessive Vitamin A intake and liver damage. [10,11]. The indirect toxicity of CAM could result from wrong dosages as well as CAM- drug interactions [12]. For instance, ginger, despite its wide use, may pose a risk of hypoglycemia when combined with diabetes medications. [11]. Furthermore, the regulations related to the CAM market are either suboptimal or absent in many countries around the world. This, coupled with the easy access to CAM products via pharmacies and retail outlets, heightens the risks that could be associated with CAM use [8,13].

At the intersection between evidence-based recommendations for CAM and sale of CAM products is the profession of Community Pharmacists (CPs) [8,14]. Drawing from their knowledge of pharmacology and their direct interaction with patients, CPs are ideally positioned to deliver evidence-based recommendations/guidance regarding CAM use [15]. More specifically, they can navigate potential risks, deliver

evidence-based information on CAM therapies, and can closely monitor patient progress. The role of CPs is further accentuated amid the prevalent non-disclosure of CAM use to physicians [6,14]. Many professional bodies have advocated for the expansion of CPs' roles to integrate standardized counselling on CAM as a core responsibility [14,16,17]. In fact, The American College of Clinical Pharmacy (ACCP) stated that pharmacist engagement with natural products is an integral part of their pharmaceutical care scope of practice [18]

The Middle East (ME) region harbors one of the fastest growing markets of CAM [19], where use of CAM modalities is common among the general populations as well as among patients with various diseases. For instance, in Saudi Arabia, CAM use was reported by 62.5% of the population and has reached over 80% among people with diabetes [20,21]. In Palestine, CAM was used by 72.8% of the general population [22], 91.5% of pregnant women [23], and 100% of children younger than 6 years [24]. The wide spread use of CAM in the ME could be attributed to the fact that, in this region, the use of natural remedies is firmly cultivated in cultural traditions and religious beliefs [25–28]. For example, Traditional Arabic and Islamic medicine (TAIM), has been practiced as a key treatment approach for millennia in the Arabian Gulf [26]. Furthermore, the political instability, and the natural and economic crisis in many countries of the region have negatively affected the structure and capacity of the conventional healthcare systems [28–30]. The fragmented and, in some instances, unavailable medical care led to an increased use of CAM as an alternative and affordable approach to address health issues [31–33].

Given the widespread use of CAM in the ME, the potential for adverse outcomes, and the lack of robust regulatory frameworks governing the CAM market, the role of CPs becomes increasingly critical. This scoping review was therefore conducted to explore and evaluate the existing evidence on the roles and responsibilities of CPs in supporting safe CAM use in the ME region.

## Methodology

In order to examine the existing evidence regarding the CPs roles in CAM in the ME region, a scoping review was conducted. This type of review was most appropriate especially considering that this research did not intend to measure any construction or provide any clinical conclusions [34]. For this purpose, the updated methodological guidance proposed by [34] for the conduct of scoping reviews was followed. This guidance consisted of 6 steps, as follows: 1) inclusion criteria. 2) search strategy 3) data extraction 4) evidence screening and selection 5) data analysis. 6) presentation of the results. This method for conducting scoping review is aligned with the PRISMA-ScR.

The research question for this review was broad to encompass a comprehensive range of insights within the subject of the investigation. Specifically, the guiding question for this review was: "What original research has been published concerning the involvement of pharmacists in Complementary and Alternative Medicine (CAM) within the ME region?".

### Inclusion criteria

For documents to be considered for inclusion in this review, they must satisfy the following inclusion criteria:

1. Published after year 2000

2. Written in languages used in ME countries namely English, French, Arabic, Turkish, Kurdish, Persian, Greek, and Hebrew

3. Be an original article or review (conference papers, books, book chapters, theses, dissertations, editorials, errata, letters, abstracts were not included.

4. Conducted partially or completely in one of the ME countries. For the purpose of this review, the Middle East was defined based on a geopolitical framework commonly adopted by the United States and NATO, encompassing the following countries: Bahrain, Cyprus, Egypt, Iran, Iraq, Israel, Jordan, Kuwait, Lebanon, Oman, Palestine (Gaza and West

Bank), Qatar, Saudi Arabia, Syria, Turkey, and the United Arab Emirates. This definition was selected to reflect both regional proximity and strategic relevance in international policy and health discourse.

5. The articles address the role of pharmacist in any form of CAM.

## Search strategy

A comprehensive literature search was conducted between July 2022 and January 2025 in 9 electronic databases MEDLINE (Ovid), Scopus, Web of Science Core Collection, Global Health, Directory of Open Access Journals (DOAJ), Al-Manhal, Arab World Research Source: Al-Masdar, E-Marefa, Iraqi Academic Scientific Journals, in addition to Google Scholar. DOAJ and Google Scholar were searched to identify any documents not indexed in the databases.

A combination of subject-specific keywords and MeSH terms were used to identify relevant literature related to the three concept domains: pharmacist, CAM, and ME countries. The search strategy was developed by the research team in consultation with a subject librarian who executed the search in the 9 databases and Google Scholar (S1 Appendix: Search Strategy). The initial search was conducted in MEDLINE (Ovid) to examine the MeSH terms and identify any related keywords or synonyms. In light of the results, the subject headings and keywords were finalized, and the search strategy was adapted for the remaining sources using Boolean operators, truncation, and proximity operators as necessary or allowed by each database. Selected databases were used to identify the Arabic keywords. The reference lists within the included articles were examined to identify any relevant documents that might have been missing. E-mail alerts were created in those databases to send notification whenever any new research related to the research question is published

The search results were exported into EndNote X9.3.3 and duplicates were removed. The de-deduplicated EndNote library was shared with reviewer 1 (HC).

## Data extraction

In alignment with the scope of this research inquiry, the research team developed a data charting form to determine the characteristics to be extracted from each research article. The form was based on previously published data extraction templates and was further adapted to address the specific objectives of this research. The data extraction form included information such as general information about each article (title, journal name, publication date, affiliation of authors, country of affiliation, type of collaboration, study design), study population, study characteristics, and pharmacy related detailed section. The full extraction form can be found in S2 Appendix: Data Charting Form. To ensure the reliability and consistency of the data extraction form, a pilot test was conducted by reviewing and extracting data from two articles using IBM SPSS.

## Evidence screening and selection

The two stages of screening were completed independently by HC and SN using EndNote. The title-abstract fields of 774 articles identified through database searching were screened and 563 documents were excluded for not meeting the inclusion criteria (93.82% agreement between the 2 reviewers). The full text of 207 articles was acquired and a detailed screening was conducted resulting in the exclusion of 76 articles (87.64% agreement between the 2 reviewers), the reasons for exclusion were: Articles published under a different name as per the record's author (n = 1), articles in which pharmacists have a minor role, focus is on other healthcare professionals (n = 34), focus is not on CAM (n = 41). Nine articles were identified during cited reference searching and 15 articles were identified through email alerts resulting in the inclusion of **155** articles in this review. The selection process is illustrated in Fig 1.

The research team conducted multiple meetings throughout the screening process to address issues related to study selection. The selection of relevant articles was a two-step process performed independently by two reviewers, namely S.N and H.C. In the initial step, the reviewers screened the titles and abstracts of the identified articles. Subsequently, the

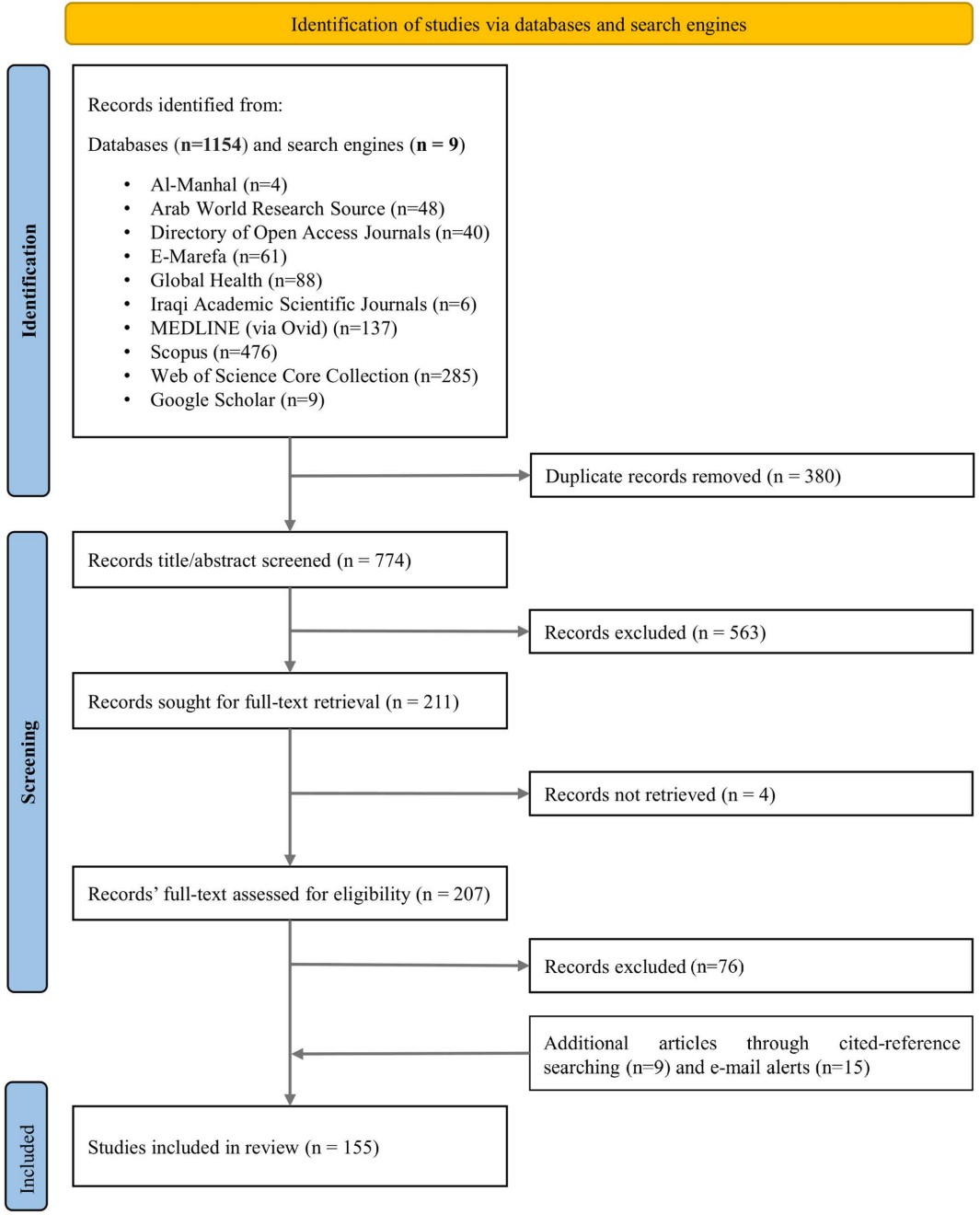

**Fig 1. The process of selecting and screening literature.** *From:* Page MJ, McKenzie JE, Bossuyt PM, Boutron I, Hoffmann TC, Mulrow CD, et al. The PRISMA 2020 statement: an updated guideline for reporting systematic reviews. BMJ 2021;372: n71. https://doi.org/10.1136/bmj.n71. For more information, visit: http://www.prisma-statement.org/.

second step involved a thorough examination of the full texts. After each step, the level of agreement between the reviewers was assessed. Disagreements concerning the inclusion or exclusion of specific documents were resolved through discussions involving the two primary reviewers, as well as consultation with two additional reviewers, F.N and M.H, who were a part of the research team. The included studies and their countries are listed in the S3 Appendix.

## Data analysis

The analysis map for this scoping review was set based on the study objectives. Thus, the analysis has considered two dimensions: the first dimension was focused on the various characteristics of the papers as per the data extraction form. The second analysis dimension was conducted to produce a broad qualitative synthesis of the outcomes stated in the collected studies. This step was accomplished by two independent reviewers, M.H. and M.A., using basic coding techniques. One of the reviewers coded the data manually, while the second coded the data with the assistance of NVivo 20. The analysis started with a pilot review of 30 random studies. During this pilot phase, coding discrepancies were discussed in detail between the two reviewers and resolved through iterative consensus-building discussions. Both reviewers independently applied initial codes to the same 30 randomly selected studies, after which their coding outputs were compared to assess consistency in interpretation and application of codes. Any disagreements were examined to refine the coding framework, clarify code definitions, and ensure a shared understanding of the thematic constructs. This process enhanced inter-rater reliability by establishing a consistent coding approach prior to analyzing the full dataset. Once alignment in coding was achieved and the refined codebook was finalized, the reviewers proceeded with coding the remaining studies independently using the agreed-upon framework. The qualitative synthesis of the data was carried out in two rounds. The first round identified the main themes while the second led to the characterization of the subthemes. To ensure the trustworthiness of the findings, themes and subthemes were validated through multiple team meetings involving both coders and senior members of the research team, during which the thematic structure was reviewed for clarity, consistency, and relevance to the research objectives.

## Results

Table 1 presents the characterization of research studies conducted to explore the role of pharmacists in CAM within the ME region during the period from 2000 to 2025. A total of 155 research documents were included in this scoping review. Among the studies included in this analysis, the majority employed cross-sectional design (98.06%) and employed quantitative methods (91.61%). Collaboration between authors from different countries was limited with the majority (57.42%) of the studies involving authors from the same country with no international collaborations. In contrast, 19.35% of the research demonstrated collaboration between ME and non-ME countries, while 10.97% involved collaboration between Middle Eastern countries. Funding sources varied among the studies. While 19.35% of the research received funding from academic sources, only 0.65% received governmental funding. A total of 75.48% of the studies indicated obtaining Institutional Review Board (IRB) approval, 23.23% did not specify their approval status, and the two reviews (1.29%) did not need ethical approvals.

Most of the studies focused on community pharmacists as participants, accounting for 53.55% of the total studies included. In terms of questionnaire validation, 60.0% of the studies reported that their questionnaires underwent a validation process, indicating a commitment to rigorous methodology. Pilot testing of questionnaires was a common practice among the studies, with 63.87% indicating that their questionnaires were subjected to pilot testing.

Fig 2 represents the time trend in the numbers of papers addressing research studies on the role of pharmacists in CAM in ME between 2000–2024 and their impact factor. The figure indicates a noticeable increase in the number of publications addressing the role of pharmacists in CAM in the ME over the years, with a significant rise observed in the most recent period (2021–2024). The mean impact factor (IF) of the publications has also shown a slight upward trend, with the highest means IF observed in the period from 2015–2020. This suggests a growing interest and potentially increasing recognition of the significance of these studies in the field of pharmacy and CAM research.

### Synthesis of the literature's findings on CPs roles in CAM in the ME region

The qualitative examination of the papers led to three main themes and several subthemes as shown in **Fig 3**, which represents the analytical process linking the overarching themes to the individual sub themes.

**Table 1. Characterization of research studies addressing role of pharmacists in CAM in ME (2000 - 2023). (n = 155).**

| Document Type | n (%) |
|---|---|
| Journal | 155(100.0) |
| **Language of publication** | **n (%)** |
| English | 151(98.9) |
| Turkish | 3(2.0) |
| Arabic | 1(1.1) |
| **Study setting** | **n (%)** |
| Saudi Arabia | 33 (21.29) |
| Jordan | 26 (16.77) |
| Turkey | 15 (9.68) |
| Palestine | 12 (7.74) |
| UAE | 11 (7.10) |
| Iran | 10 (6.45) |
| Iraq | 9 (5.81) |
| Multiple Arab Countries | 7 (4.52) |
| Lebanon | 7 (4.52) |
| Kuwait | 5 (3.23) |
| Qatar | 4 (2.58) |
| Cyprus | 3 (1.94) |
| Egypt | 3 (1.94) |
| Oman | 3 (1.94) |
| Yemen | 2 (1.29) |
| Sudan | 2 (1.29) |
| Syria | 1 (0.65) |
| A review, so NA | 2 (1.29) |
| **Study design** | **n (%)** |
| Cross-sectional | 152 (98.06%) |
| Pre-post interventional study | 1 (0.65%) |
| Reviews (systematic and scoping) | 2 (1.3%) |
| **Type of data collected** | **n (%)** |
| Quantitative | 142 (91.61%) |
| Qualitative | 7 (4.52%) |
| Mixed Methods | 4 (2.58%) |
| NA as the study design was a review | 2 (1.29%) |
| **Type of Qualitative study** | **n (%)** |
| Key-informant interview | 6 (3.87%) |
| Focus group discussion and key informant interview | 3 (1.94%) |
| Delphi technique (observational design) | 1 (0.65%) |
| **Collaboration of authors between countries** | **n (%)** |
| No collaboration, all authors same country | 89 (57.42%) |
| Collaboration with non-ME countries | 31 (20%) |
| NA, only one author | 18 (11.61%) |
| Collaboration between ME countries | 17 (10.97%) |
| **Affiliation of the authors*** | **n (%)** |
| Academic institutions | 154 (99.35%) |
| Governmental Agencies | 17 (10.97%) |
| Hospital | 15 (9.68%) |

*(Continued)*

**Table 1.** (Continued)

| Document Type | n (%) |
|---|---|
| Private Industry | 8 (5.16%) |
| NGO | 3 (1.94%) |
| Other** | 3 (1.94%) |
| **Funding sources** | **n (%)** |
| Academic | 30 (19.35%) |
| Governmental | 1 (0.65%) |
| Private | 1 (0.65%) |
| Statement of no funding | 56 (36.13%) |
| Not stated | 67 (43.23%) |
| **IRB approved among human research** | **n (%)** |
| Yes | 117 (75.48%) |
| Not Specified | 36 (23.23%) |
| NA as the study design was a review | 2 (1.29%) |
| **Type of study participants†** | **n (%)** |
| Community pharmacists | 83 (53.55%) |
| Community pharmacists only | 55 (35.48%) |
| Consumers | 33 (21.29%) |
| Hospital-based pharmacists | 44 (28.39%) |
| Physicians | 18 (11.61%) |
| Allied health professionals | 15 (9.68%) |
| Pharmacy assistants | 5 (3.23%) |
| CAM practitioners | 4 (2.58%) |
| Others | 13 (8.39%) |
| **Was the questionnaire validated?** | **n (%)** |
| Yes | 93 (60.00%) |
| Not stated | 54 (34.84%) |
| NA‡ | 6 (3.87%) |
| NA as the study design was a review | 2 (1.29%) |
| **Was the questionnaire pilot tested?** | **n (%)** |
| Yes | 99 (63.87%) |
| Not stated | 49 (31.61%) |
| NA‡ | 7 (4.52%) |
| **The main aim of the article was about CAM-Pharmacy** | **n (%)** |
| Yes | 90 (58.06%) |
| No | 65 (41.94%) |

*More than one application therefore does not add up to 100%,

**Pharmacists association, professional organization

†More than one application therefore does not add up to 100%,

‡Not applicable because a questionnaire was not used

## Pharmacists' willingness to enhance their knowledge about CAM

: Many studies elaborated on the willingness of pharmacists to improve their knowledge and skills in CAM. Many reviewed studies indicated that pharmacists have positive attitudes towards receiving education or training on CAM

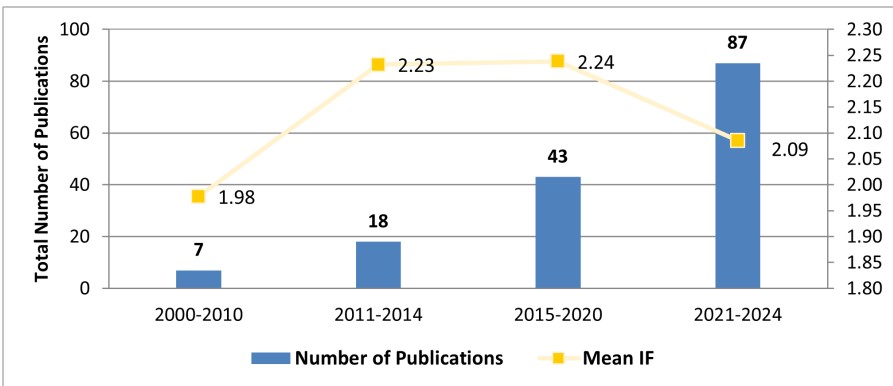

**Fig 2. Time trend in the numbers and Impact Factor (IF)\* of papers addressing research studies on the role of pharmacists in CAM in ME (2000 - 2024).** (n = 155). *Articles with no published IF according to **Journal Citation Reports** by Clarivate were omitted from IF mean calculation.

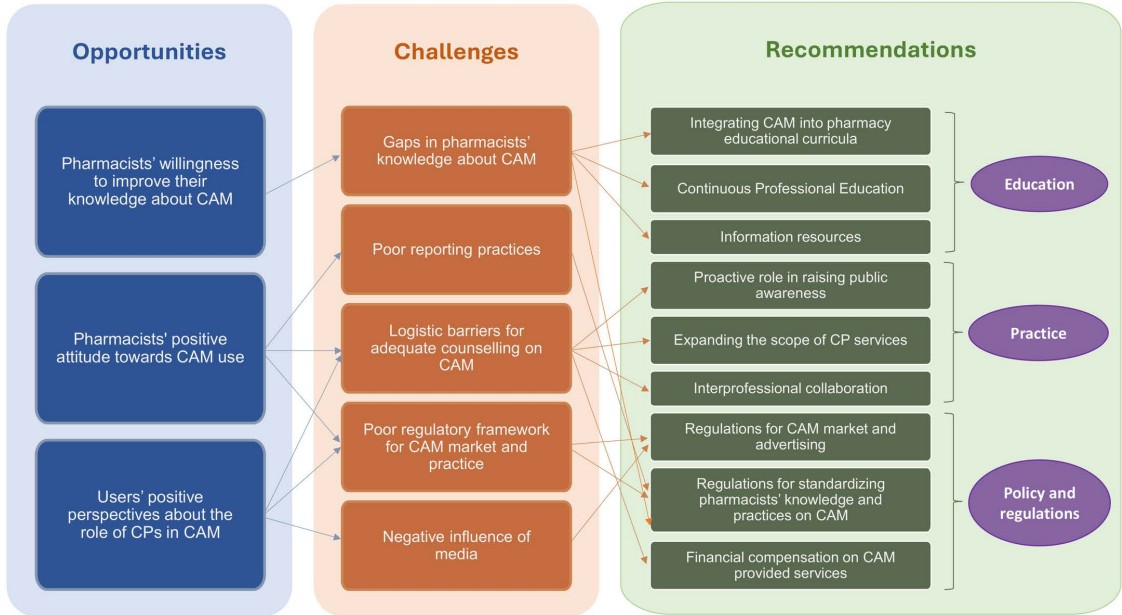

**Fig 3. Thematic map to represent the analytical process linking the overarching themes to the individual sub themes Opportunities.**

products, whether as a university requirement or as continuous education [35–42]. Additionally, several reviewed studies reported that many pharmacists are willing to refer to credible and trustworthy sources for CAM information, such as the WHO and other scientific websites [17,43–45].

**Pharmacists' positive attitude towards CAM use**

: A considerable number of reviewed studies reported that pharmacists in the ME region believe in the efficacy and safety of CAM products and that they have general positive attitudes towards CAM use [46–53]. For example, several studies found that pharmacists use CAM products themselves and recommend such products for their families and

customers [39,54–56]. Other studies reported pharmacists expressed beliefs about how CAM products could reduce problems, increase adherence, and promote the quality of life of patients [36,37,42–44,57–61].

**Users' positive perspectives about the role of CPs in CAM:** Data from the ME region showed that CAM users refer to pharmacists to obtain consultancy about CAM indications, efficacy, and safety, and that they are generally satisfied with the CAM services and consultation provided by pharmacists [47,54,58,62–76].

## Challenges

**Gaps in CPs knowledge of CAM:** Most reviewed articles found that the knowledge of CPs about CAM in the ME is inadequate, especially when it comes to side effects, adverse reactions, food and/or drug interactions, and proper dosing [39,41,44,46,48,50,52,54–56,58,61,62,77–92].

Another reported challenge was CPs' referral to unreliable scientific resources to get information about CAM. Several studies reported that the most cited CAM information sources by CPs were industry websites and representatives, product labels, or media with only a minority of CPs who reported their referral to scientific databases [55,56,80,93–97]. One study conducted in Saudi Arabia pointed out that even when CAM scientific resources were available and accessible at the practice site, CPs were not well trained to use these resources [49].

**Poor reporting practices of CAM adverse reactions or interactions:** Numerous reviewed studies indicated that CPs in the ME region inadequately identify and report adverse effects of CAM products after dispensing. Studies found that such poor reporting practices were partly due to users not disclosing their CAM use to pharmacists or other healthcare providers or never reporting encountered side effects after CAM use [41,48,78,98,99].

**Logistic barriers for adequate counseling on CAM:** CAM research in the ME region revealed various logistic and operational complexities that constrain the CPs roles in CAM, such as work overload, multiple tasks, long shifts, the absence of adequate space [89], and financial compensation. Lack of time appeared to be a common barrier for inadequate counseling on CAM [70,82,100–103]. Lack of proper communication skills and training was also a common barrier to providing proper counseling on CAM in the ME region [104–106].

**Poor regulatory framework for CAM market and practices:** Studies from various ME countries reported poor regulations of CAM market and practices. These regulations include the importation criteria, registration of CAM, quality control of CAM products, Good Manufacturing Practices of CAM manufacturers, CAM pricing and profit margin, professional responsibility of healthcare providers, including CPs and others [46,49,50,63,80,107–109]. In the same context, CPs expressed concerns about how the poor regulatory framework in the CAM field impacts their professional confidence and, as a result, limits their ability to carry out daily practice effectively across several ME countries [84,100,101,107,110].

**Negative influence of media:** A noticeable number of studies in ME discussed how invalid CAM information supplied by media (TV, magazines, social media) negatively affects the quality and credibility of CPs CAM practices [44,45,48,63,93,111–116].

## Recommendations

Recommendations from the reviewed articles were grouped into three subthemes: 1) Recommendations related to CPs practices in CAM, 2) Recommendations related to CAM policies and regulatory framework, 3) Recommendations related to CPs education on CAM.

### Recommendations related to CPs practices in CAM

**Extension of CPs services to ensure safe and effective use of CAM.**

- CPs were encouraged to provide thorough consultations on the efficacy and safety of CAM products. Moreover, pharmacists' practices on CAM were perceived as an extension to their professional role in patient care [38,47,58,60,70,89,99,100,102,103,117–121].

- Pharmacists were expected to follow evidence- based practices and not profit-oriented practices in CAM [17,48–50,54,74,80,93,96,122,123].

- Pharmacists were advised to routinely ask patients about their CAM use, follow up, and encourage users to report any adverse effects [38,48,73,79,91,98,109,124–126].

**Interprofessional collaboration.**

- CPs, physicians, nutritionists, and herbalists were encouraged to work collaboratively when providing care for CAM users [47,64,90,115,120,124,127,128].

- CPs were advised to be proactive and provide updated information on CAM to other healthcare providers [40,47,64,122,129].

**Proactive role in raising public awareness on safe and effective use CAM.** The responsibility of pharmacists to play a better role in educating the public about CAM safety, efficacy, indications, and rational use was emphasized [38,44,45,47,102,113,114,121,123,125,127,130].

## Recommendations related to CPs education on CAM

**Integrating CAM into pharmacy educational curricula.**

- Academic institutions were encouraged to integrate courses or modules on CAM use, safety, and interactions into the undergraduate curriculum for pharmacy students [35,38,39,41,48,50–52,55,56,61,62,79,81,85,86,91,93,95,97,100,101,129,131–133].

- Academic institutions were called upon to integrate training on counseling and communication skills within undergraduate curriculum for pharmacy students [70,84,92,101,103].

**Continuous Professional Education (CPE).**

- Pharmacy professional associations and groups in collaboration with academic institutions were encouraged to develop CPE programs for pharmacists that include modules and training session on CAM as well as communication and counseling skills [38,44,46,49–51,54–56,62,79,81,87–90,96,97,100,102–106,116,129,131,134].

**Information resources.**

- Pharmacy professional associations and industry management were advised to provide training for CPs on how to search for CAM information resources easily, rapidly, and effectively to make evidence-based decisions on CAM [49,56,58,94,100].

## Recommendations related to CAM policies and regulatory framework

**Regulations for CAM market.**

- Regulatory bodies and health authorities to revise and update CAM importation, registration, packaging, labeling and pricing laws [45–47,49,50,63,71,73,80,107,133,135].

- Regulatory bodies and health authorities control CAM advertising and selling outside pharmacies [44,47,108,110,111,122,136,137].

- Regulatory bodies and health authorities are encouraged to strengthen pharmacists' role by empowering them and raising public awareness of their expertise in disease treatment and prevention, using complementary and alternative

medicine (CAM), and enabling them to actively address infodemics and plan for future emergencies [17,68,76,82,90,92,109,118,120,134,138,139].

- Regulatory bodies and health authorities to establish and enforce CAM post-market surveillance [91,139], and to establish a viable reporting system including pharmacy vigilance connection network system [41,46,99].

- Regulatory bodies and health authorities establish frameworks that allow, support, and regulate the practices of pharmacists, including those related to complementary and alternative medicine (CAM), in the digital landscape, telepharmacy, and online health product marketing [131,140].

**Regulations for standardizing pharmacists' knowledge and practices on CAM.**

- Health system in collaboration with universities and professional associations to establish reliable CAM database and practice guidelines (i.e., drug and CAM information centers) to assist pharmacists in making evidence-based decisions in their daily CAM practices [49,50,106,128,131,141–143].

- Health system in collaboration with universities ensures that all pharmacists have adequate level of knowledge on CAM, and to set this as a one of the licensing requirements for CPs [47,52,92].

**Financial compensation on CAM provided services.** Regulatory bodies in collaboration with professional associations to develop and apply remuneration schemes for extended pharmacists' services in CAM advocacy and users' care [101,102].

## Discussion

The results of this review provide a detailed examination of the published literature pertaining to the landscape of the CPs' professional roles regarding CAM in the ME region. Additionally, our findings provide a thematic synthesis of the opportunities, challenges and main recommendations to enact and optimize the roles of CPs related to CAM in the ME region.

Markedly, the time trend and the number of published papers shows a contentious publication momentum which indicates a considerable growing academic and professional attention to CPs roles in CAM in the ME region. This evolving attention is in line with the global evolving interest and focus on this topic [6,97,144,145]. However, our data shows that the curve of the mean impact factor of the publications in the ME is not aligned with the curve of the publication volume. Such a lag in the impact factor could be due to recurring themes, repetitive research strategies and objectives across the examined papers. A recent systematic literature review examining the CPs' role related to CAM practices in 30 countries outside the ME region also reported a noticeable repetitive themes and ideas pattern among reviewed studies [145]. Hence the need for original and novel research undertakings, whereby researchers are encouraged to consider applying a thorough gap analysis and, then adopting innovative rigorous research strategies for addressing existing knowledge gaps in the field of pharmacy practice, including CAM [145,146].

This stagnation in journal impact factor, despite rising publication volume, may also reflect broader challenges in the region's research ecosystem, as found in this review. These include limited cross-border or international collaborations, a predominance of observational or descriptive designs with fewer interventional or longitudinal studies, and methodological shortcomings that may reduce the likelihood of acceptance in higher-impact journals. Furthermore, structural issues such as constrained research funding, limited access to research infrastructure, and underinvestment in training and capacity-building may also contribute to this trend. In addition, the review highlighted a striking lack of exploration into emerging domains such as the integration of digital health technologies in CAM pharmacy practice and the evaluation of successful policy implementation cases—both of which are essential to advancing the field in line with global healthcare innovations. Collectively, these factors may hinder the visibility, scientific influence, and citation potential of Middle Eastern research, underscoring the importance of policy-level investment in research quality, mentorship, and global engagement.

Collectively, these factors may hinder the visibility, scientific influence, and citation potential of Middle Eastern research, underscoring the importance of policy-level investment in research quality, mentorship, and global engagement.

Our findings underscore three main gaps related to the study design; first, the absence of intervention studies, which are widely acknowledged as a gold standard for producing high-quality scientific evidence in the pharmacy practice field, specifically in implementing and evaluating new programs and strategies [147,148] Second, most reviewed studies were cross sectional in nature. None of the included studies examined how the perspectives of CPs have changed over time. Furthermore, none of the studies have aimed to show a cause-effect relationship. One of the methodological recommendations to fill this important gap is to consider applying more complex longitudinal designs when more knowledge becomes available [149]. Third, less than 5% of the analyzed literature has applied qualitative or mixed methods. This limited consideration of qualitative evidence could negatively affect the depth and richness of the existing evidence, as it could contribute to the scarcity of new ideas [150,151]. Thus, it can be suggested that future research on CPs roles related to CAM in ME refers more to qualitative methods to provide a deeper understanding on what drives the behaviors of CPs and other health professionals associated with CAM. Additionally, diversifying the applied methodological tools, including qualitative research could support the generation of innovative ideas and insights to improve the research, education and practices of the field [14,145].

The findings related to the affiliation of the authors indicate that vast majority of the published research was conducted by researchers from the academic sector with modest contribution by governmental entities and significantly poor contribution by other sectors, such as industry and NGOs. While the role of academic institutions remains crucial in setting proper ethical and scientific research framework, the academic-practice or academic-business partnership in research is an important pillar for bridging the gaps in all domains of pharmacy practice (including CAM) and encouraging innovative solutions for mutually beneficial pharmacy quality enhancement [152]. Additional recommendation in this aspect is to optimize the collaboration between academic institutions and pharmacy associations (NGOs) which is seen as an important mechanism for producing real-world evidence and developing feasible CAM practice guidelines based on it [153].

Although the primary aim of scoping reviews is to map the extent and nature of existing evidence rather than to assess study quality, certain methodological concerns warrant attention. Beyond the predominance of cross-sectional designs and the lack of longitudinal and interventional studies, our review identified additional limitations in research rigor. Notably, only 60% of the studies reported using validated tools or pilot-testing their questionnaires, which raises concerns about the reliability of the data collected. Furthermore, only 75% of the studies indicated obtaining Institutional Review Board (IRB) approval, and 55% did not include any funding statement, limiting transparency and potentially introducing bias. These findings highlight the need for more robust methodological standards in future research on CAM practices in pharmacy practice.

Table 2 outlines the main gaps in the research articles reviewed in this study, as well as practical recommendation to bridge these gaps.

The thematic analysis in this scoping review shows that the reviewed studies from various countries in the ME reported similar barriers, opportunities and recommendations. Part of this similarity can be explained by the similarities in the objective, design and research tools that were applied among the studies. The consistent recommendations observed in this scoping review in various ME countries since the year of 2000 might indicate slow and intangible progress in CAM policies, education and professional development. Such a lag between evidence-based recommendations and policy development has also been reported by several studies in other regions of the world, where community pharmacists and other stakeholders are still trying to introduce policy reform to define, facilitate and expand CPs roles in CAM [14,145,153,15].

The results of this scoping review underscore the importance of standardizing CAM education and training for pharmacists in addition to improving ongoing CPE programs, as those were central recommendations across most reviewed studies. Such recommendations were prompted from the observations that current undergraduate and continuing professional training programs for pharmacists in the ME offered limited instructions on the safety, efficacy, and regulations of CAM.

**Table 2. Main gaps in the research articles reviewed in this study and suggestions for future research in the field.**

| Aspect of Research | Identified Gaps | Practical Recommendations |
|---|---|---|
| **Study design** | Predominance of cross-sectional surveys; limited intervention and cohort studies | Encourage the use of longitudinal, interventional, and mixed methods designs to assess outcomes over time and the impact of specific programs |
| | Limited qualitative research | Promote in-depth qualitative and mixed-methods research to capture contextual insights and lived experiences of pharmacists and CAM users |
| **Study tools** | Limited use of validated research instruments | Develop and adopt standardized, validated tools across studies |
| **Study themes** | Repetitive topics; lack of innovation; under-exploration of digital technologies and policy implementation outcomes | Prioritize gap analysis during research planning; promote studies that explore digital health integration, telepharmacy, AI tools, and evaluations of real-world policy implementation |
| **Collaborations** | Limited international and cross-sectoral collaborations | Strengthen inter sectoral partnerships to support multidisciplinary, cross-border projects |
| **Ethical considerations** | A sizeable proportion of studies did not report ethical approval | Require IRB approval for all studies involving human subjects as a standard practice |
| **Funding** | Constrained and inconsistently reported funding | Advocate for national and regional funding initiatives for CAM and pharmacy research; encourage greater transparency in reporting funding sources |

These findings are consistent with those reported by several other studies, which also underscored the necessity to revise and update pharmacy curriculum and accreditation requirements to guide future advancements in the pharmacy practice in the CAM field [145,153,15].

Developing and implementing well-defined practice guidelines for outlining the scope of CPs' practices associated with CAM was a recurring crucial concept across most of the analyzed studies in this review. The proposed guidelines included several priority areas, such as establishing an ethical and practical framework for promoting and selling CAM. This priority, which was signified by various recent studies in the field of CAM, was also sought as a fundamental step for improving pharmacy education and relevant resources [145,153,16,13]. Another priority area in the proposed guidelines is to set clear Adverse reactions (ADR) reporting systems to ensure safe and effective use of CAM in the ME. The same critical perquisite on CAM practices has been raised by numerous papers covering other international contexts [8,145,16,13]. These regional and global calls can be seen as a global scholarly awareness of the need for CPs to be vigilant for potential CAM ADRs and to take necessary actions if there is a substantial risk.

### Strengths and limitations

One strength of this review is that it followed the standard reporting guidelines offered by the Meta-Analysis Extension for Scoping Reviews and the Preferred Reporting Items for Systematic Reviews [34]. Another strength is using a complete double screening and reviewing technique and ensuring that every step of the procedure was conducted independently by at least two research team members. A limitation of this review is that the results were presented narratively due to the considerable heterogeneity of the analyzed literature in terms of aim, scope, population, and research methods. While narrative synthesis allowed for a broad and flexible interpretation of findings, it also carries inherent interpretive challenges. In particular, synthesizing data from studies with diverse objectives, tools, and outcome measures may limit

comparability and introduce subjective interpretation. Although potential bias was mitigated through the involvement of two independent reviewers and the use of two distinct analytic approaches (manual review and NVivo 20 software), the interpretive nature of narrative synthesis remains a source of possible bias that readers should consider. Another limitation to this review, especially to the thematic analysis section, is the fact that most of the studies identified were of cross-sectional nature. This limits the ability to draw conclusions about causality, temporal trends, or the effectiveness of interventions related to CAM practices. Future research should prioritize longitudinal, experimental, and mixed methods designs to provide deeper insights into the dynamics, outcomes, and contextual influences of CAM use. Additionally, the findings of this scoping review are inherently limited by the quality of the included studies. Methodological shortcomings—such as the limited use of validated tools, inconsistent reporting of ethical approval, and lack of funding disclosure—may affect the reliability and generalizability of the reported outcomes

## Conclusion

This review presents a comprehensive examination of the literature on the roles of CPs related to CAM in the ME region. The analysis reveals growing research activity, though characterized by repetitive approaches and limited collaboration. These findings provide a foundation for developing an evidence-informed research agenda that addresses overlooked areas, promotes innovative methodologies—including interventional, qualitative, mixed, and longitudinal designs—and fosters collaboration among academia, the pharmacy and CAM sectors, and NGOs. In the short term, policymakers and educators should integrate CAM content into pharmacy curricula, expand continuing education, and improve access to validated tools and reliable information. Long-term strategies should focus on establishing national practice guidelines, strengthening regulatory frameworks, formalizing CAM service models in pharmacies, and supporting cross-sectoral research infrastructure and global collaboration. These short- and long-term actions offer a roadmap to enhance educational, regulatory, and professional aspects of CAM pharmacy practice. The evidence presented can support researchers, policymakers, educators, and healthcare professionals in optimizing the contribution of CPs to the safe and effective use of CAM across the Middle East.

## Supporting information

**S1 Appendix. Search Strategy.**
(DOCX)

**S2 Appendix. Data Charting form.**
(DOCX)

**S3 Appendix. Included studies and their countries.**
(DOCX)

**S1 Data. PRISMA-ScR-Fillable-Checklist.**
(DOCX)

## Author contributions

**Conceptualization:** Farah Naja, Mohamad Ali Mustafa Hijazi.

**Data curation:** Maiss Ahmad, Hibeh Shatila, Sally Naalbandian, Roba Saqan.

**Formal analysis:** Maiss Ahmad, Hibeh Shatila, katia Hazim N Abu Shihab, Mohamad Ali Mustafa Hijazi.

**Methodology:** Hibeh Shatila, Sally Naalbandian, Mohamad Ali Mustafa Hijazi.

**Supervision:** Farah Naja, Mohamad Ali Mustafa Hijazi.

**Validation:** katia Hazim N Abu Shihab, Hamza Al Zubaidi.

**Writing – original draft:** Farah Naja, Maiss Ahmad, Hibeh Shatila, Mohamad Ali Mustafa Hijazi.

**Writing – review & editing:** Farah Naja, Mohamad Ali Mustafa Hijazi.

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
