## [Decision Letter · Decision Letter 0]

6 Jul 2025

Dear Dr. Hijazi,

We look forward to receiving your revised manuscript.

Kind regards,

Ali Haider Mohammed

Academic Editor

PLOS ONE

Additional Editor Comments:

1. Limited Diversity in Study Design Across Included Studies

While the scoping review is comprehensive, nearly all of the included studies employed cross-sectional designs (98.06%). This methodological homogeneity limits the depth of insight into causality, change over time, or the impact of interventions related to CAM practices. The authors should:

More explicitly acknowledge this limitation in the discussion and recommend that future studies explore longitudinal, experimental, or mixed-methods designs to advance the field.

Consider stratifying findings by study design type (e.g., observational vs. interventional) to assess whether recommendations differ depending on methodological rigor.

2. Lack of Visual Representation of Thematic Synthesis

Although the manuscript notes a thematic synthesis using NVivo, there is no visible presentation of how themes and subthemes were derived (e.g., thematic map, concept framework, or coding tree).

Please include a visual figure (e.g., thematic map or framework) showing the analytical process from raw codes to themes.

This would enhance transparency and help readers understand how conclusions were drawn from the data.

3. Absence of Quality Assessment or Risk of Bias Appraisal

Although scoping reviews typically do not require formal risk of bias assessments, the authors could improve transparency by briefly commenting on the overall methodological quality of included studies (e.g., through a descriptive overview of sample sizes, response rates, or validation processes).

Consider discussing whether any quality trends (e.g., lack of validated tools or unreported response rates) might affect the robustness of findings and recommendations.

At minimum, justify why quality appraisal was not conducted and explain how this affects interpretation.

4. Inconsistent Handling of Terminology (e.g., "Middle East")

The use of "Middle East" (ME) is not explicitly defined in terms of which countries were included. While a country list appears in the results section, it should be clearly stated earlier (e.g., in the methods or inclusion criteria).

Please define the scope of "Middle East" explicitly, possibly with reference to WHO or geopolitical classifications.

Also, consider adding a supplementary table listing the included studies by country to enhance clarity and transparency.

5. Insufficient Detail on Limitations of Narrative Synthesis

The limitations section acknowledges the use of narrative synthesis but does not elaborate on the specific interpretive challenges this presents.

The authors should clearly explain the implications of using narrative synthesis for a dataset with heterogeneous study aims, tools, and outcomes.

Acknowledge that subjective interpretation could bias the conclusions, even with dual-coding and software support.

6. Underdeveloped Discussion of Impact Factor Trends

The manuscript presents an interesting finding on the increasing volume of publications but stagnating journal impact factors. However, this point is underexplored.

The discussion would benefit from a more critical interpretation: What does this trend imply about scientific visibility, research quality, or regional research funding?

Are papers being published in lower-impact journals due to limited collaboration, lack of interventional studies, or methodological weaknesses?

7. No Explicit Link Between Themes and Policy Recommendations

The paper outlines strong thematic categories (opportunities, challenges, recommendations) but stops short of aligning these directly with practical or policy-level interventions.

Consider including a concise table that maps each theme (e.g., knowledge gaps, poor regulation) to specific, evidence-based recommendations for policy, education, or clinical practice.

Reviewers' comments:

Reviewer's Responses to Questions

**Comments to the Author**

1. Is the manuscript technically sound, and do the data support the conclusions?

Reviewer #1: Yes

2. Has the statistical analysis been performed appropriately and rigorously?

Reviewer #1: N/A

3. Have the authors made all data underlying the findings in their manuscript fully available?

Reviewer #1: Yes

4. Is the manuscript presented in an intelligible fashion and written in standard English?

Reviewer #1: Yes

Reviewer #1: General Comments

This manuscript presents a comprehensive scoping review that explores the role of community pharmacists (CPs) in the use of Complementary and Alternative Medicine (CAM) across the Middle East (ME). The topic is timely and relevant, given the widespread use of CAM and the strategic position of pharmacists to guide its use. The manuscript is well-structured, methodologically sound, and offers valuable insights into opportunities, barriers, and recommendations related to CPs and CAM. However, there are areas that need improvement to enhance clarity, coherence, and impact.

Strengths

Thorough literature search across nine databases and regional sources.

Clear thematic synthesis (opportunities, challenges, and recommendations).

Adherence to PRISMA-ScR guidelines.

Extensive reference list, suggesting strong literature coverage.

Major Comments

Novelty and Redundancy

The discussion acknowledges repeated themes across studies. While this reinforces consistency, the manuscript would benefit from more emphasis on gaps that remain under-explored (e.g., digital health integration, policy implementation success stories).

Consider adding a visual or tabular summary highlighting under-researched countries or topics within the region.

Methodology Transparency

The process of thematic analysis is briefly described, but lacks depth. Clarify how inter-rater reliability was ensured and how themes were validated.

The PRISMA flowchart is referenced but missing ("Error! Reference source not found."). Ensure figures and tables are properly inserted and labeled.

Presentation of Results

The manuscript relies heavily on textual description. Consider condensing parts of the results into summary tables (e.g., summary of challenges per country or per theme).

Some statistics are over-detailed (e.g., breakdowns of questionnaire validation); grouping similar categories could improve readability.

Language and Style

The language is generally academic, but some paragraphs are overly verbose or redundant. Example: lines 377–386 and 439–451 can be condensed without losing meaning.

Minor grammatical errors (e.g., "it is wide use" should be “its wide use”).

Policy and Practice Implications

While the recommendations are well-presented, more concrete suggestions for implementation would strengthen the practical value of the review.

It is advisable to distinguish between short-term and long-term recommendations for policymakers and educators.

Minor Comments

Line 50: Consider stating the countries covered rather than just “ME region”.

Line 78: “desire to be actively engaged in their health decision-making” – clarify or simplify phrasing.

Line 289: Correct “who reporting” to “who reported”.

Line 308: Consider rewording for clarity: “restrains their daily practices across several ME countries”.

Recommendation

Minor Revision

This manuscript is a valuable contribution to the field of pharmacy practice and CAM. With minor revisions focusing on clarity, methodological transparency, and result presentation, it can significantly enhance its impact and usefulness to readers and stakeholders.

**Do you want your identity to be public for this peer review?** For information about this choice, including consent withdrawal, please see our Privacy Policy

Reviewer #1: **Yes: ** Bassam Abdul Rasool Hassan

---

## [Author Response · Author response to Decision Letter 1]

29 Jul 2025

Dear Editor and members of the review board,

We thank you and the reviewers for the thoughtful and constructive feedback provided on our manuscript. We have carefully revised the manuscript in response to the comments and have addressed each point in detail below. The suggestions greatly contributed to improving the clarity, rigor, and relevance of our work. In the attached file, we provide a point-by-point response outlining the changes made in the revised version.

---

## [Decision Letter · Decision Letter 1]

8 Sep 2025

Role of community pharmacists in the safe and effective use of complementary and alternative medicine in the Middle East: a scoping review

PONE-D-25-28161R1

Dear Dr. Mohamad,

We’re pleased to inform you that your manuscript has been judged scientifically suitable for publication and will be formally accepted for publication once it meets all outstanding technical requirements.

Kind regards,

Ali Haider Mohammed

Academic Editor

PLOS ONE

Additional Editor Comments (optional):

Reviewer #1:

Reviewers' comments:

Reviewer's Responses to Questions

**Comments to the Author**

Reviewer #1: All comments have been addressed

2. Is the manuscript technically sound, and do the data support the conclusions?

Reviewer #1: Yes

3. Has the statistical analysis been performed appropriately and rigorously?

Reviewer #1: Yes

4. Have the authors made all data underlying the findings in their manuscript fully available?

Reviewer #1: Yes

5. Is the manuscript presented in an intelligible fashion and written in standard English?

Reviewer #1: Yes

Reviewer #1: Dear Authors,

I greatly appreciate your hard work.

All the required amendments were addressed in a very professional way.

Regards,

**Do you want your identity to be public for this peer review?** For information about this choice, including consent withdrawal, please see our Privacy Policy

Reviewer #1: **Yes: ** Bassam Abdul Rasool Hassan

---

## [Editor Report · Acceptance letter]

PONE-D-25-28161R1

PLOS ONE

Dear Dr. Hijazi,

I'm pleased to inform you that your manuscript has been deemed suitable for publication in PLOS ONE. Congratulations! Your manuscript is now being handed over to our production team.

Kind regards,

on behalf of

Dr. Ali Haider Mohammed

Academic Editor

PLOS ONE